**Open Peer Review** | Computational Biology | Methods and Protocols

# CasCADE: Cas-CRISPR Automated Design and Evaluation for targeted gRNA detection assays

Colin Price,[1] Julie L. Lucas,[1] Phil Davis,[1] Chelsey Smith,[1] Elaine Jarvis,[1] Jacqueline Fiore,[1,2] Joseph A. Russell,[1] Richard Winegar[1]

**ABSTRACT** The adaptation of CRISPR technologies for molecular detection marks a significant advancement in the field of biosurveillance and infectious disease response. CRISPR-based detection systems offer superior specificity and sensitivity compared to traditional PCR methods by directly binding and cleaving target DNA or RNA sequences, thus signaling the presence of specific pathogens. These advantages include the elimination of non-specific amplification and the reduction of required genetic material, leading to faster time to results without the need for extensive amplification cycling. However, the efficacy of CRISPR technologies heavily depends on the design of specific guide RNA (gRNA) sequences tailored for each genomic target, a process that can be intricate and time-consuming. We present Cas-CRISPR Automated Design and Evaluation (CasCADE), a state-of-the-art gRNA design software platform with a high degree of flexibility and modularity. CasCADE incorporates k-mer set operations to reduce time to answer for large data inputs when compared to computationally costly multiple sequence alignment methodologies and uses an agnostic whole genome approach to maximize gRNA discovery. CasCADE can be scaled efficiently to problems of any input sequence size and can be used for design, candidate evaluation, or both, depending on user need.

**IMPORTANCE** This work describes our software pipeline Cas-CRISPR Automated Design and Evaluation (CasCADE) that allows for *in silico* design of Cas-CRISPR detection assays. We demonstrate the viability of our design process in the lab and report 15 successful designs across nine diverse target organisms. The rapid time to answer afforded by CasCADE, combined with the superior specificity and sensitivity offered by emergent CRISPR detection assays compared to traditional PCR methods, makes this a timely contribution to pandemic preparedness and biosurveillance interests.

**KEYWORDS** CRISPR assay, detection assay, assay design, assay evaluation

CRISPR-based molecular detection systems represent a significant advancement in the field of detection assays, leveraging the specificity, programmability, and versatility of CRISPR-Cas systems to detect genetic material. Originally identified as part of the bacterial adaptive immune response (1), CRISPR systems have been adapted for a wide range of biotechnological applications, including genome editing, gene regulation, and, more recently, molecular diagnostics (2). In the context of detection assays, CRISPR technology has enabled identification of genetic material from pathogens, cancer biomarkers, or environmental targets with rapid turnaround times and high sensitivity (3, 4).

These detection systems typically utilize a CRISPR-associated (Cas) enzyme, such as Cas9, Cas12, or Cas13, in combination with a programmable guide RNA (gRNA). The gRNA comprises a spacer sequence that is complementary to the target and a scaffold region that interacts with the Cas protein. For some Cas proteins, the presence of a

**Peer Reviewer** Özlem Şahan Yapıcıer, Mehmet Akif Ersoy University, Burdur, Turkey

Address correspondence to Colin Price, cprice@mriglobal.org.

The authors declare no conflict of interest.

protospacer adjacent motif (PAM) is critical for target recognition and enzyme activation (5). Upon binding the target, Cas enzymes can either cleave the target nucleic acid or exhibit collateral activity, cleaving nearby single-stranded reporter molecules, a feature notably used by Cas12 and Cas13 enzymes (6, 7). This cleavage releases a fluorescent or colorimetric signal, making CRISPR detection assays amenable to point-of-care and field-deployable applications (8). The process in full is detailed in Fig. 1.

One of the major strengths of CRISPR-based detection is its adaptability. Assay designers may choose from an expanding range of Cas proteins with distinct properties such as RNA vs DNA targeting, PAM specificity, temperature and buffer requirements, or protein size (9, 10). As novel Cas variants continue to be discovered through metagenomic and functional screens, it is crucial that assay development frameworks remain modular (11).

The Cas-CRISPR Automated Design and Evaluation (CasCADE) platform emerges as a pioneering bioinformatics tool designed to streamline the CRISPR assay setup. It uniquely facilitates the selection and optimization of essential parameters such as inclusion and exclusion groups, Cas protein types, scaffold sequences, PAM motifs, and k-mer sizes. CasCADE enhances the process of identifying conserved sequences across selected genomic data sets, assessing them for critical properties like structural stability, free energy, and GC content, ensuring robustness in gRNA performance. This capability not only accelerates the development of robust gRNAs but also ensures their high specificity and inclusivity, essential for distinguishing target pathogens accurately. By providing rapid outputs and integrating a user-friendly approach for data handling, CasCADE represents a significant innovation in the rapid deployment of CRISPR-based detection, potentially reshaping the landscape of pathogen detection and public health preparedness (12).

## MATERIALS AND METHODS

### CasCADE pipeline

The CasCADE workflow begins by seeking to identify conserved k-mers within the target "in-group." If the in-group is a taxonomic label, programs access the National Center for Biotechnology Information (NCBI) command line data set tool to download the associated genomes. Otherwise, a manually curated genome data set with a metadata labeling table may be used instead. The size of the conserved k-mer sought for is equal to the size required by the chosen Cas protein. If the Cas protein has an associated protospacer adjacent motif region, that PAM may be specified to be required on either the 5′ or 3′ side of the conserved k-mer. This is a string-parsing approach on a per-genome record basis with set operations to analyze inclusion across the set of records and may fail especially in cases where the in-group is highly diverse and/or has a small

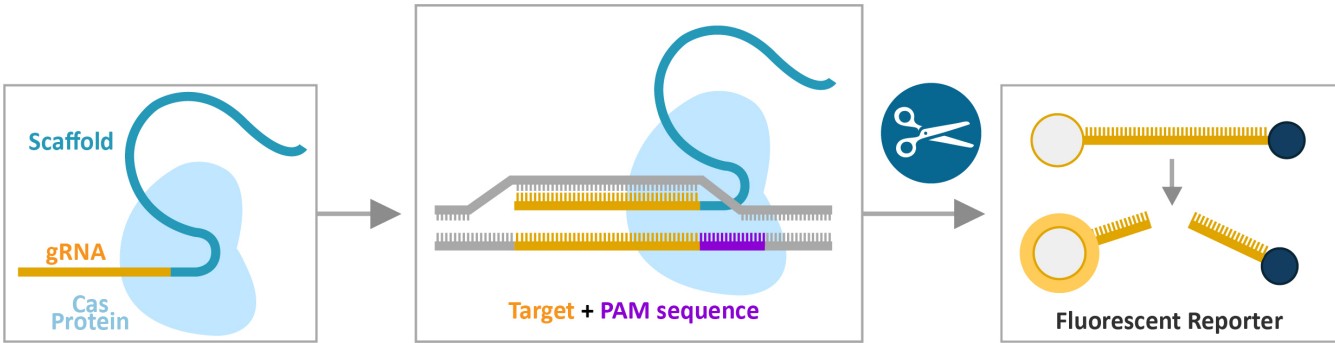

**FIG 1** Depiction of cleavage assay. Designed gRNA and scaffold sequence are integrated within Cas protein complex. The complex binds to the target as determined by the presence of the gRNA sequence and PAM sequence. This triggers cleavage of a reporter molecule with a fluorophore on one end (in white) and a quencher on the other (in black). When the reporter is cleaved, the fluorophore is sufficiently distanced from the quencher, producing fluorescence.

genome such as RNA viruses. This method of identifying regions of conservation has several advantages compared to other state-of-the-art designers that require a multiple sequence alignment (CasSilico, ADAPT) which is an NP-hard problem (13–16). The first advantage is in memory, where the iterative set operations performed on tracked k-mer counts take up space at most consisting of the string representation of the represented k-mers in a given sample. The execution of a multiple sequence alignment step requires an all-to-all comparison between sequences. This requires all memories to be loaded into memory, which quickly becomes impossible for larger data sets, for example, the set of SARS-CoV-2 sequences totaling around 273 Gb on disk, on commodity hardware. The second advantage of the k-mer set approach is the speed to answer. Multiple sequence alignments scale exponentially with the number of sequences employed, whereas k-mer set operations scale linearly. To reduce the search space size, some MSA tools will attempt to first pare down each genomic record to specific genes or other known regions of conservation (16). These regions are often chosen based on historical literature precedents, creating a self-fulfilling cycle of focusing only on small sections of genomes. This region selection has proven to be a problem in the closely related field of PCR assay design. For example, African swine fever virus (ASFV) in 2018 was spreading rapidly through Asia with a near 100% mortality rate in the pigs it infected (17). The WOAH issued a guide for molecular detection of ASFV designed initially in 2003, and in the 15 years between that initial design and the 2018 outbreak, mismatches accumulating in the p72 gene targeted by the assay resulted in PCR false negatives (18, 19). Similarly, in the 2022 Mpox outbreak, the CDC generic molecular protocol for Mpox detection used an assay designed in 2010 (20). The design targeted the TNF receptor gene. As the TNF receptor gene is a site of common deletions, this led to a failure of the assay to detect Mpox cases in September of 2022, when a deletion occurred in the target region of the TNF receptor. The assumption that previous targets will continue to work has proven to be costly, and failing to consider the entire genomic sequence can leave potentially optimal gRNA targets out.

If the initial attempt to find conserved k-mer regions fails, clustering or graph spanning operations are then employed as a second attempt; instead of finding a single perfectly conserved k-mer, a set of k-mers that are in aggregate conserved across the in-group is sought. This is done in one of two ways. The first way is through either clustering the in-group into subclusters using k-mer min-hashes generated by sourmash, and then finding a perfectly conserved k-mer in each group. The second way is a hypergraph approach, where k-mers are organized in an unweighted graph by their membership in each record and a minimum covering set is calculated that ensures every record has some representation in the output k-mer set (21). In both methods, aggregate in-group perfect inclusivity is guaranteed. In both methods, the presence of a PAM site at either the 5′ or 3′ end may be specified as required.

When candidate conserved regions have been successfully identified, the folding properties are evaluated to provide metrics to further evaluate designs. The scaffold sequence is added on to the k-mer from the previous step at either the 5′ or 3′ end, depending on the properties of the Cas protein before evaluation as a collective candidate gRNA. Properties calculated include the GC content, the scaffold free energy, the scaffold folding structure, the gRNA free energy, and the gRNA folding structure. From these metrics, gRNAs are filtered down to the top candidates by taking the most stable gRNA free energy value, filtering out gRNAs without preserved scaffold structures, and considering only gRNAs within a specified GC content range. This process is detailed in Fig. 2. To keep the problem computationally tractable in the interests of a 24-hour full assay design period, should many gRNAs pass these filtering operations, they may be randomly uniformly sampled to a subset of 24 candidate gRNAs to evaluate further for exclusivity.

After filtering, a label is attached to the flat tabular data file output from the filtering operations containing the TaxId label or custom labeling scheme used to define the target in-group. From here, two data groups are retrieved to evaluate for inclusivity,

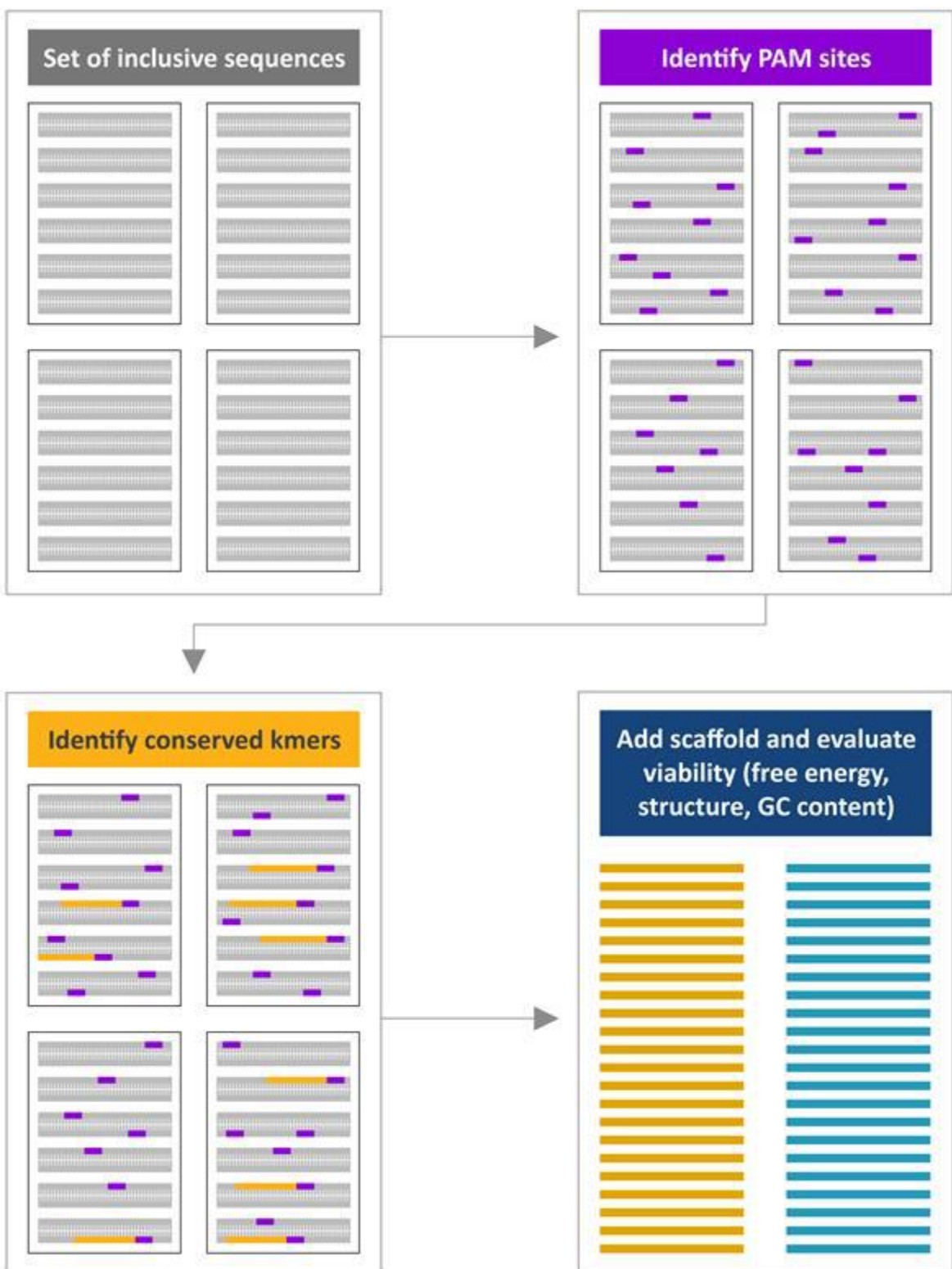

**FIG 2** Depiction of candidate generation and evaluation. First, if required by the Cas protein type, PAM sequences are identified across each sequence file. Conserved regions that co-occur next to the identified PAM sites are identified through inner set joins across input records. If no PAM is required, all conserved k-mers are considered for design. Identified k-mers are then concatenated to the scaffold sequence where free energy, structural stability, and GC content are evaluated and filtered onto a final candidate guide RNA set.

which contains the set of all records belonging to the in-group, and exclusivity, which contains the set of all records in the above taxonomic tree-level record. For example, if an assay is species level, the exclusive group is defined as the genus for that species. These retrieval processes are automated in the taxonomic label case using NCBI Datasets. To evaluate inclusivity, a BLAST database is made of the inclusive record, and each candidate gRNA is BLASTed. BLAST results are parsed to make sure they match the length of the gRNA exactly and contain no more than a single mismatched nucleotide. If a PAM was specified, it is parsed for in the hit record to ensure it is present on the appropriate 5′ or 3′ side. The output is a tabular report of the percent inclusivity, which ought to be 100% as the previous steps guarantee inclusivity, as well as the total number of hits in the chance that a gRNA viably hits a genomic record more than one time.

Similarly, for exclusivity, a BLAST database is made from the exclusive record set, and each candidate gRNA is BLASTed. BLAST results are parsed for matches of the length of the gRNA with no more than one mismatch. If a custom labeling was specified, the labels of the in-group are not considered as "off-target" hits, while everything else within this mismatch criteria is considered an off-target hit. A hit to a genome in the exclusive group is not considered an off-target hit in the case that a PAM is specified and the PAM does not occur at the required 5′ or 3′ side. The output is two tabular reports of the number of off-target hits, one reporting in terms of accession and one in terms of the taxonomic name of the off-target hit. This process is repeatable for the NCBI's NT database that contains all nucleotide records if evaluation against a broader exclusivity group is desirable. Finally, a consideration for the gRNAs to hit the human genome as an off-target is evaluated. The GRCh38 database is BLASTed against using the same criteria as the previous exclusive checks. If a human is found to be hit by the gRNA and PAM sequence, it is indicated in the tabular output as a Boolean "true" hit. A gRNA that is perfectly inclusive, exclusive to taxonomic near neighbors, and exclusive to human signal is considered an ideal candidate to be used in the detection assay. The end-to-end pipeline is detailed in Fig. 3.

## CasCADE gRNA designs

The viability of CasCADE-designed gRNA assays was passed through CasCADE's *in silico* inclusivity and exclusivity analysis as well as laboratory chemistry validation. Nine different targets are presented here: *Abrus precatorius* (abrin toxin gene), pan-*Candida* genus, *Burkholderia mallei/pseudomallei*, *Klebsiella aerogenes*, *Klebsiella pneumoniae*, *Vibrio vulnificus*, *Plasmodium falciparum*, *Pseudomonas aeruginosa*, and *Ricinus communis* (preproricin gene). For the two toxin targets, *Abrus precatorius* and *Ricinus communis*, inclusive sequence records were retrieved using NCBI's BLAST web portal. A total of 14 *Abrus precatorius* and 51 *Ricinus communis* were retrieved for design. The other targets were retrieved using the NCBI Datasets download tool, specifying only records indicated to be complete. Sixty-eight *Burkholderia mallei/pseudomallei*, 99 *Klebsiella aerogenes*, 529 *Klebsiella pneumoniae*, 54 *Vibrio vulnificus*, 18 *Plasmodium falciparum*, and 1,583 *Pseudomonas aeruginosa* records were retrieved. Exclusivity analysis was performed against all other species at the genus level for species-level targets and at the genus level for the one *Candida* pan-genus target. All exclusive records were retrieved using the NCBI Datasets download tool. The retrieved records for design were then used to evaluate each of the nine batches of designs for inclusivity. For *Abrus precatorius*, *Burkholderia mallei/pseudomallei*, *Plasmodium falciparum*, *Vibrio vulnificus*, *Ricinus communis*, and pan-*Candida* genus, perfect exclusivity was found for the reported designs. For *Klebsiella aerogenes* and *Klebsiella pneumoniae*, off-target hits were found in other *Klebsiella* genus members. For *Pseudomonas aeruginosa*, one off-target hit to *Pseudomonas aeruginosa* was found.

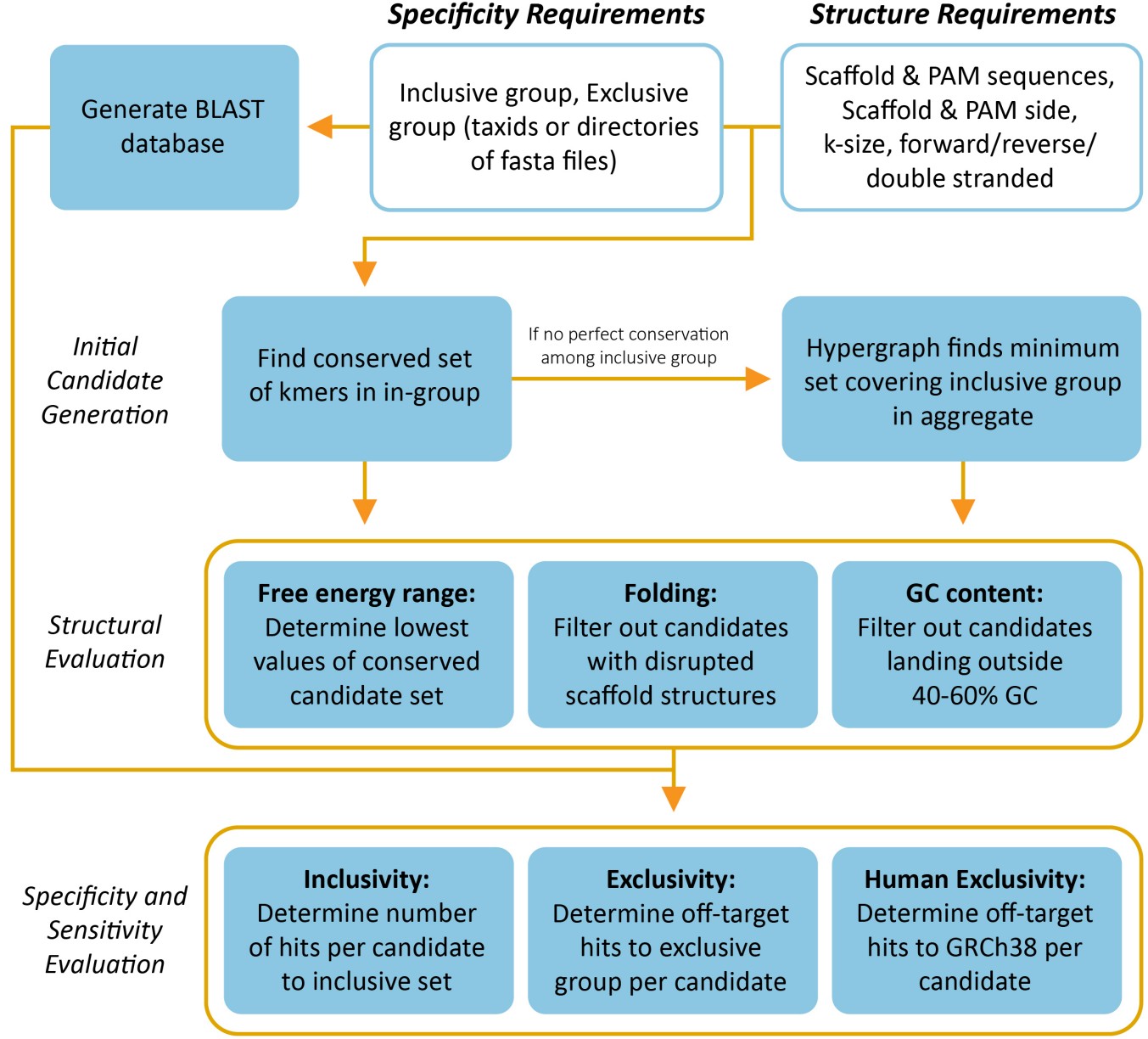

**FIG 3** CasCADE complete process diagram. CasCADE begins with specification of an input inclusive group of sequences to target, Cas protein parameters, and optionally an exclusive group of sequences to avoid. Initial candidates are generated from identifying conserved k-mers across all inclusive records. If sequences are not conserved enough to find total conservation, a hypergraph method is employed to find a set of conserved regions that in aggregate are conserved across all records. Candidates must also occur next to a PAM site if specified as a Cas protein parameter. Candidates are then evaluated to identify candidates with preserved scaffold structures, optimal GC content, and the lowest free energy. Surviving candidates are then evaluated for inclusivity to the inclusive group, exclusivity to the exclusive group, and exclusivity to the human GRCh38 database using BLAST. Off-target hits are considered hits if the query sequence is one or fewer mismatches from the mapped-to sequence.

## Solution-based CRISPR Cas12 detection

### Materials

All targets were acquired from ATCC or BEI Resources and extracted using the RNeasy PowerFecal Pro Kit (Qiagen) per manufacturer's instructions. Once extracted, targets were quantified using the Qubit 1× dsDNA High Sensitivity (Thermo Fisher) and stored in −80°C until use. All gRNAs designed from the CasCADE pipeline were purchased from Integrated DNA Technologies (IDT) with HPLC purification. The Cas12 enzyme and

Cas12 buffer were purchased from Molecular Cloning Laboratories. A Cas12 single-stranded reporter was purchased from IDT: /56-FAM/TTATTATT/3IABkFQ/. MicroAm Optical 384-Well Reaction Plate with Barcode (Applied Biosystems) was used as the plate. All measurements were made using a QuantStudio 5 384-well Real-Time PCR System (Applied Biosystems).

### Solution-based CRISPR

Direct detection was used to determine guide activity using a solution-based CRISPR protocol. The following were combined in one tube to create the Cas enzyme complex: 40 nM of the gRNA to be tested, 20 nM active Cas12 enzyme, and RNase inhibitor (NEB) in 1× Cas12 buffer. The Cas enzyme complex was mixed and incubated at 37°C for 30 minutes. After incubation, 400 nM of ssDNA reporter was added and mixed. The Cas enzyme complex was subsequently placed on ice until ready to use.

Targets were diluted in water and RNase inhibitor to the desired concentrations. First, 15 µL of complexing reaction was added to each well, followed by 5 µL of the correct target dilution or water. The plate was sealed and centrifuged for 1 minute before transferring to the QuantStudio 5. The plate was incubated at 37°C for 2 hours, with a read every 5 minutes in the FAM channel.

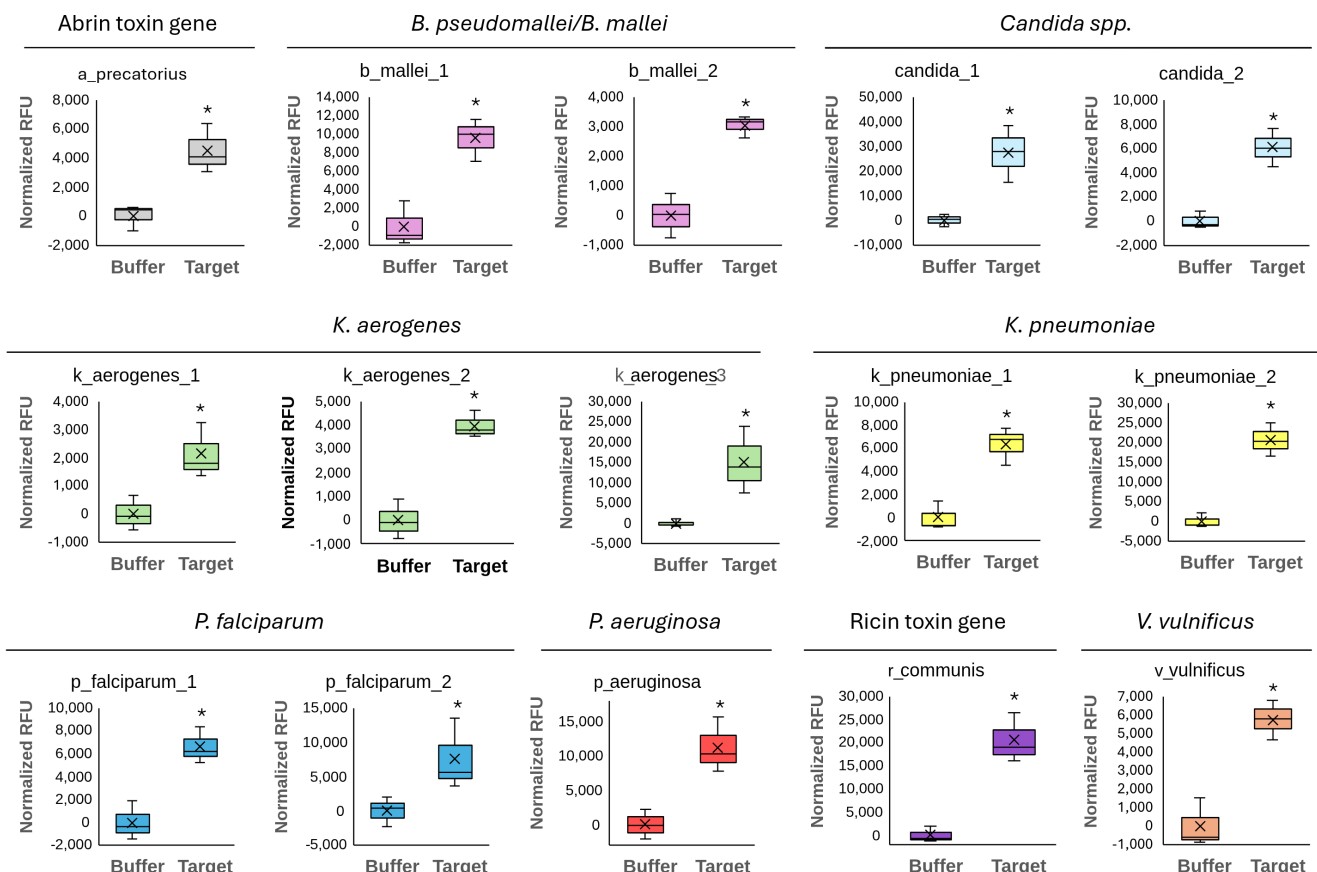

**FIG 4** Activity of CasCADE-designed gRNAs. We designed and tested 15 gRNAs representing nine different targets: *Abrus precatorius* (abrin toxin gene, in gray), *Burkholderia mallei/pseudomallei* (in pink), pan-*Candida* genus (in light blue), *Klebsiella aerogenes* (in green), *Klebsiella pneumoniae* (in yellow), *Plasmodium falciparum* (in dark blue), *Pseudomonas aeruginosa* (in red), *Ricinus communis* (preproricin gene, in purple), and *Vibrio vulnificus* (in orange). Each target was tested in triplicate, and statistical significance was determined using the Wilcoxon rank-sum test. *$P \leq 0.05$.

## RESULTS

### Design and validation

To demonstrate the practical viability of CasCADE designs, we present 15 gRNA designs for nine distinct targets that have each been experimentally validated. These assays include *Abrus precatorius* (abrin toxin gene), pan-*Candida* genus, *Burkholderia mallei/pseudomallei*, *Klebsiella aerogenes*, *Klebsiella pneumoniae*, *Vibrio vulnificus*, *Plasmodium falciparum*, *Pseudomonas aeruginosa*, and *Ricinus communis* (preproricin gene). These targets form a diverse set, notably representing fungi, bacteria (including GC-rich), parasites, and plants. The target, number of inclusive sequences used in designing, the guide sequence, and the guide plus the scaffold sequence are detailed in Table S1. The designed assays were verified using the described solution-based CRISPR chemistry in the Materials and Methods section and Fig. 1. For this study, we used a direct detection assay, without preamplification of the target. Target preamplification (usually with isothermal chemistry such as LAMP or RPA) is generally used to improve sensitivity of CRISPR-based assays. This simplified the study design and allowed for an accurate evaluation of guide activity not confounded by varying efficiencies of amplification. Each guide was tested for the assay target and a negative buffer control in triplicate. Average normalized RFU values for all 15 guides are plotted in Fig. 4. All targets demonstrated elevated RFU compared to background levels, with a clear separation between buffer and target. To assess the statistical significance of detection, the Wilcoxon rank sum test (non-parametric) was to compare RFU values of each target against its corresponding background control. All 15 targets yielded statistically significant differences ($P \leq 0.05$), confirming consistent signal elevation above baseline noise. Average RFU ranged from 2,128 (k_aerogenes_1) to 27,330 (candida_1). Additionally, candida_1 and k_pneumoniae_2 assays were evaluated for human exclusivity and were both found to be significant by Wilcoxon rank-sum test. A comparison between the tested target, buffer, and human A549 cells is shown in Fig. 5.

## DISCUSSION

The deployment of the CasCADE platform represents a significant advancement in the design of guide RNAs for diverse biological targets, including viruses, fungi, bacteria, plants, and animals. To date, CasCADE has facilitated the generation of hundreds of candidate gRNAs *in silico* across a diverse target set that are ready for future in-lab

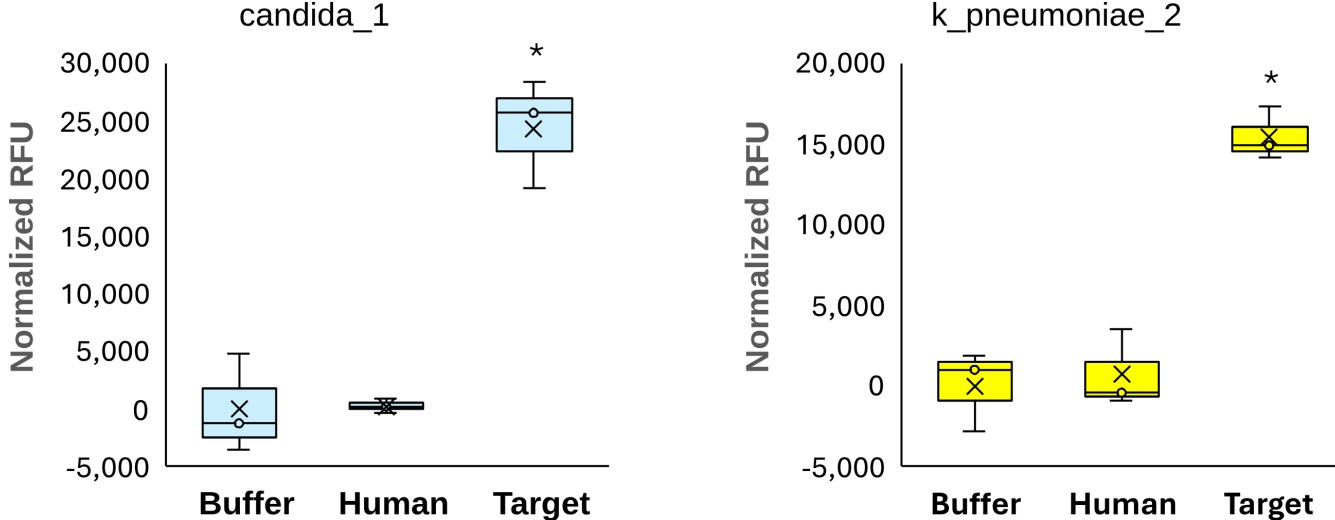

**FIG 5** Human exclusivity of CasCADE-designed gRNAs. We performed an additional human exclusivity evaluation for two gRNAs, one targeting pan-*Candida* and one targeting *Klebsiella pneumoniae* against human A549 lung cells. Each target was tested in triplicate, and statistical significance was determined using the Wilcoxon rank-sum test. *$P \leq 0.05$.

validation, underpinning broad-spectrum pathogen detection capabilities. The ongoing efforts at MRIGlobal to validate these gRNAs *in vitro* and refine scoring methods to refine the selection process reflect the commitment to optimizing CasCADE's efficacy. As more data are accumulated on the performance of these gRNAs, MRIGlobal will harness machine learning techniques to predict *in vitro* gRNA success more accurately. This predictive model will leverage the vast array of data generated by CasCADE, aiming to refine gRNA designs further based on empirical outcomes. Additionally, the modular nature of the CasCADE pipeline allows for the seamless integration of new design criteria or adjustments to existing parameters, ensuring that the platform remains adaptable to emerging scientific insights and detection needs.

In instances where it is challenging to identify perfectly conserved gRNA designs, alternative strategies such as the hypergraph method are employed. This approach, along with decision tree-based methods and the incorporation of degenerate IUPAC characters for scenarios like single-nucleotide polymorphism detection, ensures comprehensive coverage and inclusivity of the target spectra. Continued refinement of these approaches and development of new techniques to address these "corner cases" in assay design will expand the capabilities of CasCADE further.

As generated candidate gRNAs continue to be validated *in vitro*, they will offer critical insights that could drive further iterations of CasCADE, enhancing its precision and reliability in molecular detection. This iterative process of validation and refinement is essential not only for optimizing the technical parameters of the CasCADE platform but also for ensuring that the gRNAs it produces can reliably meet the stringent requirements of modern biosurveillance and diagnostic applications.

## ACKNOWLEDGMENTS

Support from the DARPA Biological Technologies Office as part of the Detect It with Gene Editing Technologies program funded under the Naval Information Warfare Center contract N66001-21-1-4048, which is awarded to MRIGlobal. The authors thank Craig Willis, Pamela Winegar, Landon Adebiyi, and Sarah Pope for programmatic support.

The views, opinions, and/or findings expressed are those of the authors and should not be interpreted as representing the official views or policies of the Department of Defense or the U.S. Government. Approved for public release: distribution is unlimited.

## AUTHOR AFFILIATIONS

[1]MRIGlobal, Kansas City, Missouri, USA
[2]MedStar Georgetown University Hospital, Washington, DC, USA

## AUTHOR ORCIDs

Colin Price  http://orcid.org/0000-0001-9769-5125
Julie L. Lucas  http://orcid.org/0000-0001-6094-1740
Chelsey Smith  http://orcid.org/0000-0002-9341-0580
Joseph A. Russell  http://orcid.org/0000-0002-0623-5519

## DATA AVAILABILITY

The code used to run the CasCADE pipeline is available at https://github.com/mriglobal/CasCADE. All output guide designs are published in the Table S1. All sequences used in the design are freely available through NCBI's data sets tools and were retrieved through use of the TaxID identifier.

## ADDITIONAL FILES

The following material is available online.

## Supplemental Material

**Table S1 (Spectrum02920-25-s0001.xlsx).** Supplementary guide designs.

## Open Peer Review

**PEER REVIEW HISTORY (review-history.pdf).** An accounting of the reviewer comments and feedback.

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
