## [Reviewer comments · Microbiology Spectrum]

Microbiology Spectrum

CasCADE – Cas CRISPR Automated Design and Evaluation for Detection Assay Design

Colin Price, Julie Lucas, Phillip Davis, Chelsey Smith, Elaine Bradford, Jacqueline Fiore, Joseph Russell, and Richard Winegar

Corresponding Author(s): Colin Price, MRIGlobal

Review Timeline:

Submission Date:	September 12, 2025
Editorial Decision:	November 24, 2025
Revision Received:	January 22, 2026
Accepted:	February 4, 2026

Editor: Max Maurin

Reviewer(s): Disclosure of reviewer identity is with reference to reviewer comments included in decision letter(s). The following individuals involved in review of your submission have agreed to reveal their identity: Özlem Şahan Yapıcıer (Reviewer #2)

Transaction Report:

DOI: <https://doi.org/10.1128/spectrum.02920-25>

Re: Spectrum02920-25 (**CasCADE - Cas CRISPR Automated Design and Evaluation for Detection Assay Design**)

Dear Dr. Colin Price:

Thank you for the privilege of reviewing your work. Below you will find my comments, instructions from the Spectrum editorial office, and the reviewer comments.

Revision Guidelines

Sincerely,
Max Maurin
Editor
Microbiology Spectrum

Reviewer #1 (Comments for the Author):

The authors of the submitted work named CasCADE - Cas CRISPR Automated Design and Evaluation for Detection Assay Design - have touch with their work an unmet need in the field of CRISPR-Cas based detection methods. Their work is well written and of high interest to the field. There are two major issues in this work that requires attention. The first one is to show that the guideRNAs and thus the assay will not pick up genetic material of other bacteria. A simple blast search shows that some of the guideRNAs also give a 100% hit with other bacterial species, e.g. the guideRNAs of *K. pneumoniae* also give a 100% hit with *Escherichia coli*, *Citrobacter*, *Yersinia*, *Salmonella* to name a few. I think it is of high importance to show whether

the CRISPR-Cas test will also test positive or negative for the genetic material of these closer related bacteria and thus provide false positive results. If this is the case the authors should mention that in their work and discuss these limitations so that the authors or the field can work on improving these limitations and take this into account when using their method. Moreover, i also think it is important to use these guideRNAs on genomic DNA of the human host. Although i agree that their work indeed shows that it is unlikely that the host will become positive it is only in silico data, confirmation with wet lab work will strengthen their paper in my opinion.

Minor comments:

CRISPR-Cas only detects genetic material; the diagnosis is made by the physician. Although the authors highlight this so now and then in the manuscript, i would suggest to adapt the sentences where the claims are made of CRISPR-Cas diagnostics toward the detection of genetic material, of which the result outcome(s) might be used by the physician to make a diagnosis.

Secondly, if the authors want to use the name CasCADE also for commercial purposes i would suggest to adapt this name, since it is already used for a Type I CRISPR-Cas system e.g., in *E. coli* and as a genome editing tool. This would help to prevent confusion and other issues.

Reviewer #2 (Comments for the Author):

Dear Authors,

Thank you for your study.

The study is strong, original, and scientifically valuable, particularly for a methods-oriented journal in bioinformatics, synthetic biology, CRISPR diagnostics, or pathogen detection.

Sincerely,

Responses due Saturday January 23, 2027

Reviewer #1 (Comments for the Author):

The authors of the submitted work named CasCADE - Cas CRISPR Automated Design and Evaluation for Detection Assay Design - have touch with their work an unmet need in the field of CRISPR-Cas based detection methods. Their work is well written and of high interest to the field. There are two major issues in this work that requires attention. The first one is to show that the guideRNAs and thus the assay will not pick up genetic material of other bacteria. A simple blast search shows that some of the guideRNAs also give a 100% hit with other bacterial species, e.g. the guideRNAs of *K. pneumoniae* also give a 100% hit with *Escherichia coli*, *Citrobacter*, *Yersinia*, *Salmonella* to name a few. I think it is of high importance to show whether the CRISPR-Cas test will also test positive or negative for the genetic material of these closer related bacteria and thus provide false positive results. If this is the case the authors should mention that in their work and discuss these limitations so that the authors or the field can work on improving these limitations and take this into account when using their method.

Running blastgRNAexclusive on the guides on a database of all bacteria could be done to satisfy questions about greater exclusivity. Running blast alone is insufficient as it does not consider the PAM sequence. Our claims for exclusivity evaluation were done for just "near neighbors" in the genus, or the family level for Pan-genus candida assays. We do comment on our exclusivity findings in these sections:

"Exclusivity analysis was performed against all other species at the genus level for species level targets and at the genus level for the one *Candida* pan-genus target."

"For *Abrus precatorius*, *Burkholderia mallei/ pseudomallei*, *Plasmodium falciparum*, *Vibrio vulnificus*, *Ricinus communis*, and pan-*Candida* genus, perfect exclusivity was found for reported designs. For *Klebsiella aerogenes* and *Klebsiella pneumoniae*, off-target hits were found to other *Klebsiella* genus members. For *Pseudomonas aeruginosa*, one off-target hit to *Pseudomonas aeruginosa* was found."

Moreover, i also think it is important to use these guideRNAs on genomic DNA of the human host. Although i agree that their work indeed shows that it is unlikely that the host will become positive it is only in silico data, confirmation with wet lab work will strengthen their paper in my opinion.

We were able to perform some in-lab analysis on human material to validate our exclusivity approach further on a couple of assays, which is shown in new Figure 5. If we had more time and resources to commit, we are confident that we would validate the remainder.

Minor comments:

CRISPR-Cas only detects genetic material; the diagnosis is made by the physician. Although the authors highlight this so now and then in the manuscript, i would suggest to adapt the sentences were the claims are made of CRISPR-Cas diagnostics toward the detection of genetic material, of which the result outcome(s) might be used by the physician to make a diagnosis.

We have updated the language surrounding "diagnosis" and "diagnostic" to "detection" in cases where explicitly describing how CasCADE is used currently. We aspire through continued refinement to have the technologies behind CasCADE work in diagnostic settings.

Secondly, if the authors want to use the name CasCADE also for commercial purposes i would suggest to adapt this name, since it is already used for a Type I CRISPR-Cas system e.g., in *E. coli* and as a genome editing tool. This would help to prevent confusion and other issues.

We've decided to keep CasCADE for the sake of the current publication but thank the reviewer for bringing this to our attention. We will reconsider the name in the future.

Reviewer #2 (Comments for the Author):

Dear Authors,
Thank you for your study.

The study is strong, original, and scientifically valuable, particularly for a methods-oriented journal in bioinformatics, synthetic biology, CRISPR diagnostics, or pathogen detection.
Thank you for taking the time to review our manuscript.

Re: Spectrum02920-25R1 (**CasCADE - Cas CRISPR Automated Design and Evaluation for Detection Assay Design**)

Dear Dr. Colin Price:

Your manuscript has been accepted, and I am forwarding it to the ASM production staff for publication. Your paper will first be checked to make sure all elements meet the technical requirements. ASM staff will contact you if anything needs to be revised before copyediting and production can begin. Otherwise, you will be notified when your proofs are ready to be viewed.

Sincerely,
Max Maurin
Editor
Microbiology Spectrum

Reviewer #1 (Comments for the Author):

The manuscript has been further improved, but I still have some minor remarks that require attention. In the method section a statistical analysis section is missing, which needs to be added as in line with the articles published in the spectrum journal.

In this sentence after "was" the word "used" is missing "To assess the statistical significance of detection, the Wilcoxon rank sum test (non-parametric) was to compare RFU values of each target against its corresponding background control." I would suggest to check the manuscript thoroughly on such minor errors further.

Moreover, although the manuscript is improved the limitations of PAM presence are not well discussed. This is also what I was referring to concerning that experiments were needed to prove that a 100% guideRNA match with a target without a PAM can still result in cutting, e.g. when the DNA is in a bubble structure. I suggest instead of an experiment to further include this in the discussion with a couple of sentences to highlight the limitations of using the PAM motif in the present work to inform the readers on this subject, so that they will be aware that in some cases false positive detection results can be generated, because of such limitations.